

# A comparison of observation-level random effect and Beta-Binomial models for modelling overdispersion in Binomial data in ecology & evolution

Xavier A. Harrison

Institute of Zoology, Zoological Society of London, UK

## ABSTRACT

Overdispersion is a common feature of models of biological data, but researchers often fail to model the excess variation driving the overdispersion, resulting in biased parameter estimates and standard errors. Quantifying and modeling overdispersion when it is present is therefore critical for robust biological inference. One means to account for overdispersion is to add an observation-level random effect (OLRE) to a model, where each data point receives a unique level of a random effect that can absorb the extra-parametric variation in the data. Although some studies have investigated the utility of OLRE to model overdispersion in Poisson count data, studies doing so for Binomial proportion data are scarce. Here I use a simulation approach to investigate the ability of both OLRE models and Beta-Binomial models to recover unbiased parameter estimates in mixed effects models of Binomial data under various degrees of overdispersion. In addition, as ecologists often fit random intercept terms to models when the random effect sample size is low (<5 levels), I investigate the performance of both model types under a range of random effect sample sizes when overdispersion is present. Simulation results revealed that the efficacy of OLRE depends on the process that generated the overdispersion; OLRE failed to cope with overdispersion generated from a Beta-Binomial mixture model, leading to biased slope and intercept estimates, but performed well for overdispersion generated by adding random noise to the linear predictor. Comparison of parameter estimates from an OLRE model with those from its corresponding Beta-Binomial model readily identified when OLRE were performing poorly due to disagreement between effect sizes, and this strategy should be employed whenever OLRE are used for Binomial data to assess their reliability. Beta-Binomial models performed well across all contexts, but showed a tendency to underestimate effect sizes when modelling non-Beta-Binomial data. Finally, both OLRE and Beta-Binomial models performed poorly when models contained <5 levels of the random intercept term, especially for estimating variance components, and this effect appeared independent of total sample size. These results suggest that OLRE are a useful tool for modelling overdispersion in Binomial data, but that they do not perform well in all circumstances and researchers should take care to verify the robustness of parameter estimates of OLRE models.

Corresponding author
Xavier A. Harrison,
x.harrison@ucl.ac.uk

## INTRODUCTION

Binomial data are frequently encountered in the fields of ecology and evolution. Researchers often wish to know what factors determine the proportion of offspring sired by a focal individual (*Tyler et al., 2013*), the proportion of eggs of a clutch that successfully hatch (*Harrison et al., 2013a*), or the prevalence of disease in a population (*Bielby et al., 2014*). To determine which factors drive variation in the proportion data of interest, researchers often fit Binomial models to their data and model the Binomial mean as a function of covariates. However, in many cases these Binomial models exhibit overdispersion, where the variance of the data is greater than that predicted by the model (e.g., *Zuur et al., 2009*; *Bolker et al., 2009*). Failing to deal with overdispersion can lead to biased parameter estimates and standard errors in these models (*Hilbe, 2011*; *Harrison, 2014*), potentially leading to false conclusions regarding which covariates are truly influential on the outcome variable. It is therefore crucial that we find robust means to deal with overdispersion in order to correctly identify the biological processes underlying our observed Binomial data.

Several methods to deal with overdispersion are currently available. As overdispersion can downwardly bias standard errors in models, one method involves 'correcting' the standard errors by multiplying them by the square root of the dispersion coefficient (*Zuur et al., 2009*). This multiplicative correction for overdispersion occurs when one specifies the 'quasi' family in Generalized Linear Models (GLMs) in the statistical software R (*R Core Team, 2014*). However, a weakness of the 'quasi' approach is that it does not model the overdispersion in the data, but merely adjusts the resulting parameter estimates with a single correction factor. The assumption that all standard errors are biased to the same degree is an obvious problem, which may not be appropriate (e.g., *Harrison, 2014*, Table 1). The alternatives to the 'quasi' approach for proportion data are to explicitly model the source of extra-Binomial variation in the data (e.g., *Williams, 1982*; *Hughes & Madden, 1993*; *Lee & Nelder, 1996*; *Richards, 2008*), for example by using compound probability structures (e.g., Beta-Binomial models), or to use observation-level random effects (OLRE). With OLRE models, each observation in the model receives a unique level of a random effect that absorbs the extra-Binomial variation in the data, hopefully yielding a model with unbiased parameter estimates and without overdispersion. However, although several studies have sought to investigate the utility of OLRE to model overdispersion in Poisson count data (*Elston et al., 2001*; *Harrison, 2014*), similar investigations for Binomial proportion data are relatively rare. *Harrison (2014)* found that for Poisson data, OLRE yielded accurate parameter estimates and $r^2$ values in most situations of overdispersion, but that OLRE could not adequately cope with overdispersion caused by zero-inflation. Here I will address the shortfall in our understanding of the capacity of OLRE to model overdispersion in Binomial data, with a specific focus on mixed effects models. In order for OLRE to be an appropriate tool, they should be robust to the process generating overdispersion in the data, and thus I test OLRE on overdispersed Binomial data generated by a variety of mechanisms. In addition, I explore the utility of Beta-Binomial hierarchical models as an alternative to OLRE models, and compare the accuracy of parameter estimates derived from both approaches.

## A typical Binomial example

For these examples, I will assume the outcome variable that we are measuring is the number of eggs $h$ that have hatched out of a total clutch $c$ for a hypothetical lizard species. I assume that the proportion of hatched eggs is described by a Binomial distribution:

$$h_i \sim Binomial(c_i, p_i) \tag{1}$$

where $h_i$ is the number of eggs hatched by individual $i$ from its total clutch $c_i$, with mean probability $p_i$. In a typical Binomial model, we can model the mean hatch rate $p_i$ as a function of covariates of interest. Let us assume that hatch rate shows a positive relationship with how many prey items an individual lizard has consumed, and also that there is a weak negative relationship between body size and hatch rate. Let us also assume we have measured $N$ individuals from $J$ populations of lizards, and that we wish to control for variation among populations using a random intercept.

$$logit(p_i) = alpha_{j(i)} + \beta_{prey} \times Prey_i + \beta_{bodysize} \times Bodysize_i \tag{2}$$

$$alpha_j \sim Normal(\mu_{pop}, \sigma^2_{pop}) \quad \text{for } j = 1, \ldots, J, \tag{3}$$

$$p_i = \frac{1}{(1 + \exp(-logit.p_i))} \tag{4}$$

where the hatch rate $p_i$ is a function of a linear model. $alpha_{j(i)}$ is the intercept for population $j$ to which individual $i$ belongs, where each $alpha_j$ is drawn from a normal distribution with mean $\mu_{pop}$ and variance $\sigma^2_{pop}$ (Eq. (3)). $\beta_{prey}$ and $\beta_{bodysize}$ are the slope parameters for the effects of number of prey items consumed and body size, respectively. $Prey_i$ and $Bodysize_i$ are the prey and body size measurements of individual $i$. We convert the linear predictor ($logit.p_i$ from, Eq. (1)) back to a probability ($p_i$) using a *logit* link (Eq. (4)). Suppose now we wanted to model our hatching success data using these covariates. Statistical packages such as *lme4* (*Bates et al., 2014*) readily fit such generalized linear mixed models (GLMMs)

```
m1<-glmer(cbind(hatch,clutch-hatch)~ Prey
        + Bodysize + (1|Population),family=binomial(logit))
                                                    R Code
```

where hatch and clutch are vectors where each row corresponds to the measurements for a single individual $i$ for $h_i$ and $c_i$ from Eq. (1). Prey and Bodysize are vectors of measurements of $Prey_i$ and $Bodysize_i$ corresponding to Eq. (2). Population is a vector denoting the population ID of each individual. We also specify the Binomial error distribution with a *logit* link using the 'family' argument in the *glmer* call. When we fit model *m1*, we are modeling our data according to Eq. (1)–Eq. (4) above. The model will estimate $\beta_{prey}$, $\beta_{bodysize}$, $\mu_{pop}$, and $\sigma_{pop}$. In order to be confident that the resulting parameter estimates are robust, we should check for overdispersion in model *m1*. *Bolker et al. (2009)* and *Harrison (2014)* provide R code to calculate the dispersion parameter for such models. Briefly, a point estimate of the dispersion parameter can be calculated as the ratio of the sum of
squared Pearson residuals to the residual degrees of freedom for the model, where a value >1 indicates overdispersion (*Zuur et al., 2009*; *Bolker et al., 2009*). *Harrison (2014)* also provides code to estimate the dispersion parameter and 95% confidence intervals using parametric bootstrapping. If the data exhibit overdispersion, we can adjust our model to take this into account, either with observation-level random effects, or using a compound error structure such as the Beta-Binomial.

## Modeling overdispersion using observation-level random effects

Including an observation-level random effect requires that we modify Eq. (2) above to include an additional term in the linear predictor:

$$logit(p_i) = alpha_{j(i)} + \beta_{1prey} \times Prey_i + \beta_{bodysize} \times Bodysize_i + \varepsilon_i \tag{5}$$

$$\varepsilon_i \sim Normal(0, \sigma_\varepsilon^2) \tag{6}$$

where $\varepsilon_i$ is an additional term unique to each observation $i$ that is drawn from a normal distribution with a mean of 0 and variance $\sigma_\varepsilon^2$. If a dataframe D containing the observations has $N$ rows, we can create an observation level random effect as follows:

```
obs<-seq(nrow(D))
```
*R Code*

where D is the dataframe in which the values of Prey and Bodysize are stored. We can then modify our model *m1* to include the OLRE denoted by 'obs':

```
M2<-glmer(cbind(hatch,clutch-hatch)~ Prey
    + Bodysize + (1|Population + (1|obs),family=binomial(logit))
```
*R Code*

Model *m2* will estimate the same parameters as *m1*, but in addition will also estimate the additional parameter $\sigma_\varepsilon^2$. The larger the value of $\sigma_\varepsilon^2$, the greater the degree of overdispersion in the dataset. The magnitude of the variance parameter $\sigma_\varepsilon^2$ can be informative, for example when compared to hierarchical variance components (e.g., individual nested within brood, nested within site (see *Elston et al., 2001*)). However, in many cases the OLRE will simply 'soak up' the extra-Binomial variation in the data, effectively treating the overdispersion as nuisance variation. The problem with this approach is that often the overdispersion might be biologically interesting (*Zuur et al., 2009*) and indeed relevant to our hypotheses regarding the processes underlying variation in the observed data. An alternative way to model overdispersion is by using hierarchical models such as Beta-Binomial models.

## Modeling overdispersion using hierarchical Beta-Binomial models

An alternative to adding an observation-level random effect to models involves modelling the overdispersion using compound probability distributions such as the Beta-Binomial. The benefit of this approach is that by quantifying the process generating the overdispersion (through the estimate of $\phi$, see below), one may gain a more precise understanding of the ecological mechanisms underlying observed data (*Martin et al., 2005*; *Richards,*

*2008*). For Beta-Binomial models, the linear predictor remains the same as Eq. (2)–Eq. (4) above, on the *logit* scale but instead of drawing observed counts directly from a Binomial distribution with mean $p_i$, we draw the Binomial probabilities from a beta distribution with parameters $a$ and $b$:

$$beta.p_i \sim Beta(a_i, b_i) \tag{7}$$

$$a_i = \frac{p_i}{\phi} \tag{8}$$

$$b_i = \frac{(1 - p_i)}{\phi} \tag{9}$$

$$h_i \sim Binomial(c_i, beta.p_i) \tag{10}$$

where $a_i$ and $b_i$ are the shape and scale parameters of the Beta distribution for individual $i$, calculated using the value of $p_i$ (Eq. (4)) and $\phi$, which is the constant overdispersion term in the model. As with $\sigma_\varepsilon^2$ for the OLRE models above, the larger the value of $\phi$ the greater the degree of overdispersion in the data.

## Overdispersion and sample size

Multiple features biological data can influence the accuracy with which models recover parameter estimates for effect sizes. Overdispersion is likely a ubiquitous feature of the kinds of 'messy' data collected by ecologists from field and laboratory studies, and is known to bias parameter estimates in count data (*Hilbe, 2011*; *Harrison, 2014*). In addition, overdispersion can arise for a variety of reasons, including aggregation (heterogeneity) in the data, or through failing to measure important covariates, or include relevant interactions between covariates in models (*Hilbe, 2011*). However, multiple factors may interact with overdispersion to add further bias to models of overdispersed data, including most notably the sample size of the datasets. To date, relatively little is known in the ecological literature about the interaction between overdispersion and sample size and how this affects parameter estimates. This is particularly relevant to mixed effects models, where the number of grouping levels of a random intercept term (e.g., number of populations) can greatly influence model accuracy. Low replication at the level of the random effect grouping variable can mean there is not enough information to estimate the variance among groups, especially if one employs 5 or fewer levels (*Gelman & Hill, 2006*, p. 247). Unfortunately, ecologists often fit factors containing fewer than 5 levels as random effects (e.g., a random intercept for 'Year,' *Harrison et al., 2013a*; *Harrison et al., 2013b*), largely because gathering 3 or 4 years of data in the laboratory or field represents an enormous amount of work.

In order for OLRE to be considered a robust tool for modeling overdispersion in Binomial data, they should yield accurate parameter estimates under a broad range of conditions, including high overdispersion and low sample size. This paper investigates the influence of 3 specific variables on the accuracy of parameter estimates from mixed models: (i) for a fixed sample size, the influence of weak, moderate and strong overdispersion; (ii) for strong overdispersion, the influence of the level of replication of the random intercept

term; and (iii) for strong overdispersion, the influence of Binomial sample size. For all three scenarios, I consider overdispersion resulting from two mechanisms, using either a Beta-Binomial distribution or an overdispersed Binomial distribution to generate the data (see equations above). Finally, I use both OLRE and Beta-Binomial models to assess the relative performance of each model type for a given scenario of overdispersion and sample size. Model performance is assessed by (i) quantifying the accuracy with which the models can recover estimates of $\beta_{prey}$, $\mu_{pop}$ and $\sigma^2_{pop}$ (values fixed for all simulations); and (ii) including a weak negative effect of $\beta_{bodysize}$ ($-0.01$), corresponding to roughly a 4% difference in reproductive success between the smallest and largest individuals in the dataset, and quantifying the proportion of simulation replicates for a given scenario that incorrectly inferred a positive slope for $\beta_{bodysize}$. Such an outcome is important, because most ecological datasets likely contain variables of weak effect that are 'biologically relevant' to the organism(s) in question, but our ability to detect such effects in the presence of overdispersion has received relatively little attention (but see *Richards, 2008*).

## METHODS

### Data generation

I explored the consequences varying three key parameters in Binomial mixed models: (i) the magnitude of overdispersion ('Overdispersion' Scenario), (ii) the number of the levels of the random intercept term, ('Random Effect' Scenario) and (iii) the Binomial sample size (number of trials per observation) ('Binomial Sample Size' Scenario). For each scenario, I simulated data from both an overdispersed Binomial distribution using Eq. (1)–Eq. (6) and a Beta-Binomial distribution using Eq. (1)–Eq. (4) and Eq. (7)–Eq. (10) (see Introduction) to examine whether the accuracy of the mixed models also depended on the mechanism generating the overdispersion in the data. For the Overdispersion simulations, $\phi$ was set at 0.1, 1, or 2 for the Beta-Binomial data, and $\sigma_\varepsilon$ (specified as standard deviation, not variance $\sigma^2_\varepsilon$, in the R code) set to 0.1, 1.5 or 3 for the corresponding overdispersed Binomial data. A value of $\sigma_\varepsilon = \phi = 0.1$ corresponds to weak overdispersion (model dispersion parameter ~1.1), whereas $\sigma_\varepsilon = 3 / \phi = 2$ corresponds to a model dispersion parameter of ~2. For all simulations parameter values were fixed at the following: $\mu_{pop} = -1$; $\sigma_{pop} = 0.5$; $\beta_{bodysize} = -0.01$; $\beta_{prey} = 0.6$. For the Overdispersion and Binomial Sample Size scenarios, I assumed 10 different populations had been sampled, each with a sample size of 20 individuals ($n = 200$). For the Random Effect scenario, the number of populations was set at 3, 5 or 20. Clutch size ($C_i$) was fixed at 5 for the Overdispersion and Random Effect simulations, but was set to 2, 4 or 10 for the Binomial Sample Size scenario. Full details of the parameters used in each of the 3 scenarios are provided in Table 1.

### Model fitting simulations

All simulations were coded in R v3.1.1 (*R Core Team, 2014*). One thousand datasets were simulated for each set of three different parameter estimates for each of the three scenarios and data types (Beta-Binomial or overdispersed Binomial, see Table 1). For each dataset, I

**Table 1 Parameter values for simulation scenarios employed in the study.** '$\phi$,' overdispersion parameter for Beta-Binomial models; '$\sigma_\varepsilon$,' overdispersion parameter for overdispersed Binomial models; '$n$ trials,' Binomial sample size (maximum clutch) for simulations, equivalent to $C_i$ in Eq. (1); '$n$ individuals,' number of individuals per simulated population; '$n$ populations,' number of populations simulated for each dataset, and fitted as a random intercept term in all models, referred to as random effect sample size. Values for the variable under investigation in each scenario are shown in bold.

| | Overdispersion $\phi/\sigma_\varepsilon$ | $n$ trials (clutch size) | $n$ individuals | $n$ populations |
|---|---|---|---|---|
| **Overdispersion** | | | | |
| 1 | **0.1/0.1** | 5 | 20 | 10 |
| 2 | **1/1.5** | 5 | 20 | 10 |
| 3 | **2/3** | 5 | 20 | 10 |
| **Levels of random effect** | | | | |
| 1 | 2/3 | 5 | 20 | **3** |
| 2 | 2/3 | 5 | 20 | **5** |
| 3 | 2/3 | 5 | 20 | **20** |
| **Binomial sample size** | | | | |
| 1 | 2/3 | **2** | 20 | 10 |
| 2 | 2/3 | **4** | 20 | 10 |
| 3 | 2/3 | **10** | 20 | 10 |

fitted model m2 containing an OLRE (see Introduction) in the *lme4* package and extracted parameter estimates for $\mu_{pop}$, $\sigma_{pop}$, $\beta_{prey}$, $\beta_{bodysize}$ and $\sigma_\varepsilon$ (the SD of the observation level random effect, '*obs*'). Following 1,000 simulations, I calculated simulation means and 95% quantiles for parameters. I also calculated the proportion of models that falsely estimated the effect of body size to be positive ($\beta_{prey} > 0$). Data for the proportion of models where $\beta_{prey} > 0$ are presented as means and bootstrapped 95% confidence intervals for each parameter/data type combination. I did not test for significant differences between mean values for each parameter.

For each of the three scenarios and two data types, I fitted a corresponding Bayesian Beta-Binomial hierarchical model in JAGS (*Plummer, 2013*) using the R package *runjags* (*Denwood, 2014*), following Eq. (1)–Eq. (4) and Eq. (7)–Eq. (10) above. This resulted in four combinations of data-generating process and statistical model used in analysis: Beta-Binomial data with OLRE, overdispersed Binomial data with OLRE, Beta-Binomial data with a Beta-Binomial Model, and overdispersed Binomial data with Beta-Binomial model. The Bayesian framework is extremely flexible, meaning models following these equations can be easily specified, even though few frequentist mixed model packages in R permit the fitting of Beta-Binomial models (but see *spaMM*, *Rousset & Ferdy, 2014*; *glmmADMB*, *Fournier et al., 2012*). Models were run for 20,000 iterations with a thinning interval of 20 following a burnin of 2,000. Convergence was assessed by running two parallel chains and calculating the Gelman–Rubin statistic, which was below 1.05 for parameters, indicating convergence. Results are presented as posterior means and 95% credible intervals for all parameters. I used uninformative Normal priors with mean 0

and precision 0.001 for $\mu_{pop}$, $\beta_{prey}$ and $\beta_{bodysize}$; an uninformative uniform prior on the interval (0,10) for $\sigma_{pop}$, and an uninformative gamma prior with $a = b = 0.001$ for $\phi$. To test sensitivity of model output to choice of priors, I reran models where $\phi$ had a uniform prior on the interval (0,10) and $\sigma_{pop}$ had a gamma prior where $a = b = 0.001$. Results from both sets of models were similar, suggesting limited sensitivity to prior specification.

Unlike frequentist models, their Bayesian equivalents are much more computationally intensive and thus slower to run. Because of this, I generated only 1 dataset and ran 1 model for each scenario/data type combination ($n = 18$). Both Bayesian and frequentist data are plotted alongside one another in the figures, but it is important to note that the frequentist data are the 95% intervals of the distribution of 1,000 means for each parameter, whilst the Bayesian data are the 95% credible intervals of 1,000 samples from the parameter space for a single mean. Although they are slightly different quantities, the point of the comparison is to assess the relative accuracy of a Beta-Binomial model compared to a Binomial model allowing for overdispersion on the linear predictor i.e., containing observation-level random effects. However because of the use of Bayesian analyses in these simulations, the type of model (OLRE or Beta-Binomial) is therefore confounded with the fitting algorithm (Maximum Likelihood or Bayesian, respectively). That is, frequentist methods may perform poorly in generalized mixed models (*Ferkingstad & Rue, 2015*), and Bayesian methods may perform slightly better, and this may have little to do with the type of model. To test the sensitivity of the parameter estimates to Beta-Binomial modeling philosophy, I reran the data simulations for highly overdispersed Binomial ($\sigma_\varepsilon = 3$) and Beta-Binomial ($\phi = 2$) data with 10 populations, 20 individuals per population and a clutch size of 5 per individual. Instead of Bayesian Beta-Binomial models, I fitted frequentist Beta-Binomial mixed models using the 'spaMM' package' and extracted means and 95% confidence intervals for parameters after 1,000 simulations.

Model code for the Bayesian models, and all data simulations in the manuscript are provided in Online Supplementary Information.

## RESULTS

### Overdispersion

Weak overdispersion ($\sigma_\varepsilon / \phi = 0.1$) resulted in accurate parameter estimates for $\beta_{prey}$, $\mu_{pop}$ and $\sigma_{pop}$ for all four data/models combinations as expected (Fig. 1). However, for both moderate and strong overdispersion, bias increased for all parameters when the data were generated from a Beta-Binomial distribution but analysed using OLRE (yellow circles, Fig. 1). Conversely, the overdispersed Binomial/OLRE model did not suffer the same bias, although the standard error of all estimates increased in tandem with overdispersion (blue circles, Fig. 1). Beta-Binomial models performed well for both Beta-Binomial and overdispersed Binomial data (yellow and blue diamonds, Fig. 1), but were unable to accurately estimate $\sigma_{pop}$ when overdispersion was high ($\sigma_\varepsilon / \phi = 3/2$ respectively). Increasing overdispersion caused an increase in the proportion of models incorrectly inferring a positive slope for $\beta_{bodysize}$ for both types of data (Fig. 4A). *Summary: OLRE are highly sensitive to the mechanism generating the overdispersion in the data, yielding*

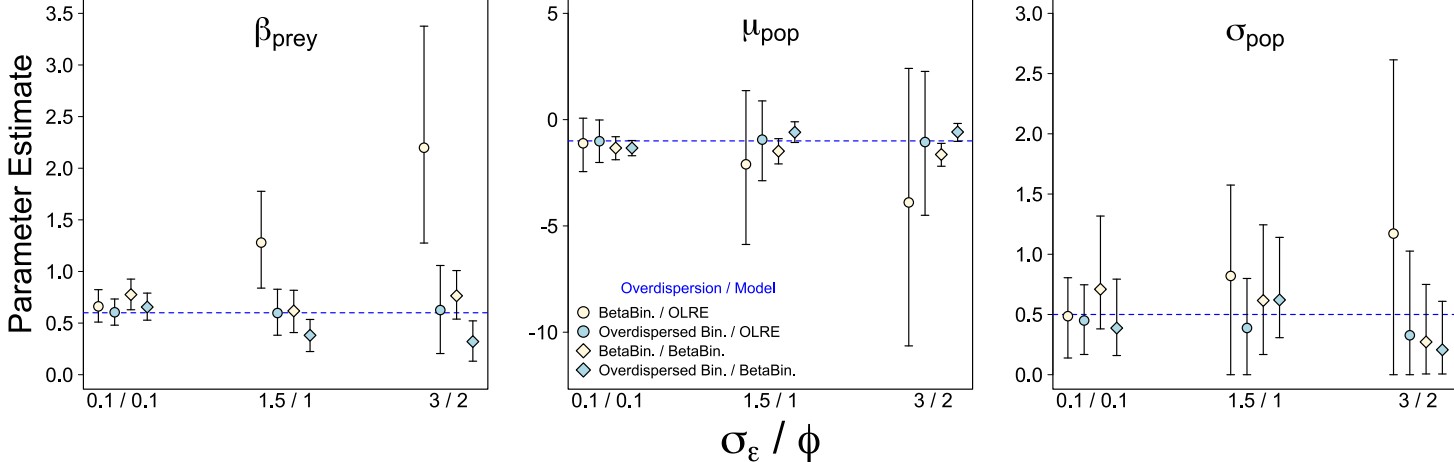

**Figure 1** **Effect of varying degrees of overdispersion on parameter estimation.** Parameter estimates and 95% intervals for 3 levels of overdispersion under 4 combinations of overdispersion and model type. Yellow circles, Beta-Binomial overdispersion and an observation-level random effect (OLRE) model; blue circles, overdispersed Binomial data and OLRE model; yellow diamonds, Beta-Binomial data and a Beta-Binomial model; blue diamonds, overdispersed Binomial data and a Beta-Binomial model. Beta-Binomial data were analysed using Bayesian Hierarchical Beta-Binomial mixed models and so error bars are 95% credible intervals. '$\beta_{prey}$,' slope parameter for effect of number of prey items consumed; '$\mu_{pop}$,' mean value of population random intercept term; '$\sigma_{pop}$,' standard deviation of population random effect. X axis labels refer to the overdispersion parameters for each model type: overdispersed Binomial models, $\sigma_{\varepsilon}$, the standard deviation of a random effect with mean 0 added to the linear predictor; Beta-Binomial models, $\phi$, the dispersion parameter for the Beta-Binomial mixture distribution.

*large bias when applied to Beta-Binomial data. Parameter bias gets progressively worse as overdispersion increases.*

## Number of levels of the random effect

For all four data/model combinations, the precision of the estimates increased as the number of levels of the random effect increased from 3 to 20 (Fig. 2). This was expected, as a higher number of levels yields more information to estimate hierarchical modeling components such as $\sigma_{pop}$. However, for the Beta-Binomial data/OLRE model, increasing the number of levels yielded both a consistently biased mean value for $\beta_{prey}$, and increased precision around the biased mean (yellow circles, Fig. 2). OLRE models on overdispersed Binomial data (blue circles) performed generally well for both $\beta_{prey}$ and $\mu_{pop}$. Conversely a Beta-Binomial model on overdispersed Binomial data (blue diamonds, Fig. 2) tended to underestimate $\beta_{prey}$. For all four data/model combinations, estimates of $\sigma_{pop}$ were highly imprecise when only 3 populations were considered. This was especially true for both Beta-Binomial models (yellow and blue diamonds, Fig. 2). There was still a large degree of bias when $n = 5$, especially when OLRE models were applied to Beta-Binomial data (yellow circles and diamonds). The proportion of models recovering $\beta_{bodysize} > 0$ tended to decrease as the $n$ increased, but only by 5% on average, and there appeared to be no differences in proportion depending on the kind of overdispersion generated in the data. Increasing the sample size to $\sim$200 but using only 3 populations (67 individuals per population) still resulted in biased parameter estimates, especially for Beta-Binomial models (Table 2A), suggesting it is random effects sample size and not total sample size driving this pattern. *Summary*: *Higher replication of the random effects results in more*

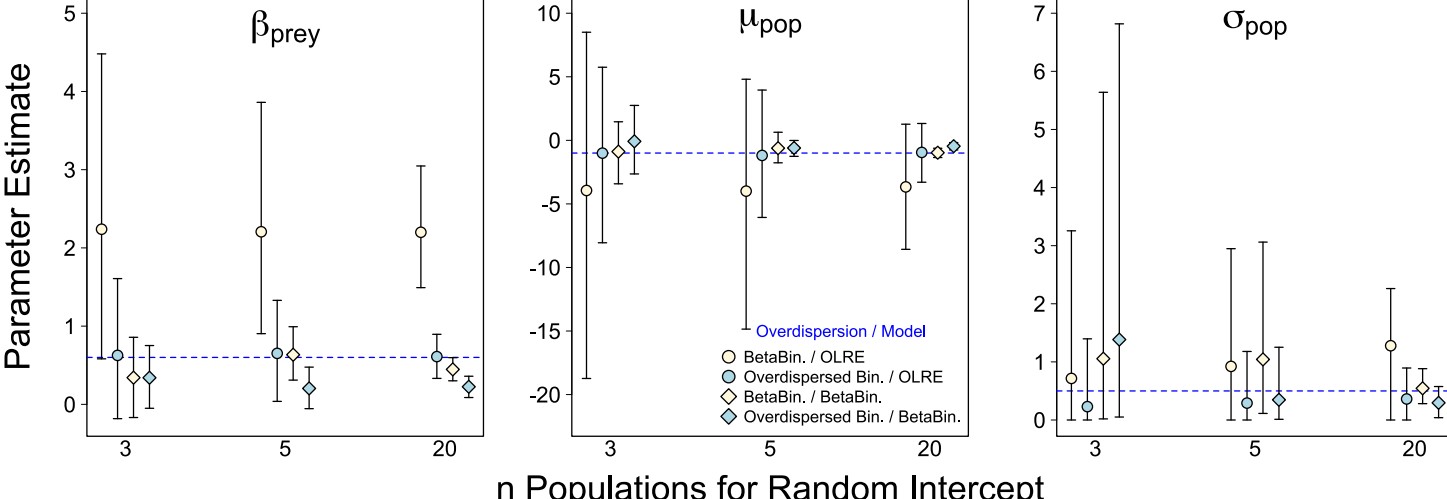

**Figure 2** **Effect of varying sample size of the random intercept term (number of populations) on parameter estimates in the presence of overdispersion.** Parameter estimates and 95% intervals for 3 different levels of replication of the random intercept term under 4 combinations of overdispersion and model type. Yellow circles, Beta-Binomial overdispersion and an observation-level random effect (OLRE) model; blue circles, overdispersed Binomial data and OLRE model; yellow diamonds, Beta-Binomial data and a Beta-Binomial model; blue diamonds, overdispersed Binomial data and a Beta-Binomial model. Beta-Binomial data were analysed using Bayesian Hierarchical Beta-Binomial mixed models and so error bars are 95% credible intervals. '$\beta_{prey}$,' slope parameter for effect of number of prey items consumed; '$\mu_{pop}$,' mean value of population random intercept term; '$\sigma_{pop}$,' standard deviation of population random effect. For all simulations $\sigma_\varepsilon$, was set to 3 for overdispersed Binomial models, and $\phi$ set to 2 for Beta-Binomial models.

precise estimates, but cannot fix the bias caused by overdispersion when using OLRE for some data types. All models perform poorly when the number of populations is 3, especially for $\sigma_{pop}$, suggesting there is no modeling 'fix' for poor replication of the random effect.

### Binomial sample size

Of the three scenarios considered, Binomial sample size had the smallest effect on parameter accuracy and precision. As with the Random Effects scenario, increasing Binomial sample size did not remedy the bias in $\beta_{prey}$ caused by overdispersion, but did yield slightly higher precision (yellow circles, Fig. 3). Estimates for all other parameters were similar irrespective of Binomial sample size and type of overdispersion (blue circles, yellow diamonds and blue diamonds, Fig. 3), and the proportion of models where $\beta_{bodysize} > 0$ were fairly constant across all values tested (Fig. 4C). *Summary: OLRE models on Beta-Binomial data continue to perform poorly in the presence of overdispersion, irrespective of the Binomial sample size. All other models performed equally well, and there was evidence suggesting that even over the narrow range of sample sizes tested (2–10), precision increased with sample size.*

### A comparison of frequentist and Bayesian beta-binomial estimates

Model estimates from the Beta-Binomial models were very similar irrespective of whether they were from frequentist or Bayesian models (Fig. 1 and Table 2B). Interestingly, both the frequentist and Bayesian models suggested that using Beta-Binomial models

**Table 2 Model results investigating the effects of total sample size and model type on parameter estimates.** (A) Parameter values and 95% confidence intervals for 1,000 simulations of data where 3 populations were simulated each with a sample size of 67 ($n = 201$) and analysed with OLRE models. Results are highly similar to when 3 populations are simulated each with a sample size of 20 ($n = 60$, see Fig. 2), suggesting it is replication of the random effects and not total sample size driving the poor performance of models. True simulated values are shown in parentheses. 'Beta-Binomial Data' refer to data generated from a Beta-Binomial mixture model with dispersion parameter $\phi = 2$; 'overdispersed Binomial Data' refers to data generated by adding random noise to the linear predictor of a Binomial model on the link scale, from a Normal distribution with mean 0 and standard deviation $\sigma_\varepsilon = 3$. (B) Parameter values and 95% confidence intervals for 1,000 simulations of overdispersed data analysed with frequentist Beta-Binomial models. Results are highly similar to their Bayesian equivalents (see Fig. 1), suggesting it is the mechanism generating the overdispersion in the data that results in poor parameter estimates and not modelling philosophy (frequentist/maximum likelihood vs. Bayesian).

| | Beta-Binomial data | | Overdispersed binomial data | |
|---|---|---|---|---|
| Parameter | Mean | 95% CI | Mean | 95% CI |
| **(A)** | | | | |
| $\beta_{prey}$ (0.6) | 2.2 | [1.33, 3.41] | 0.61 | [0.19, 1.07] |
| $\beta_{bodysize}$ ($-0.01$) | $-0.03$ | [$-0.34$, 0.29] | $-0.007$ | [$-0.18$, 0.17] |
| $\mu_{pop}$ ($-1$) | $-3.82$ | [$-11.79$, 2.99] | $-1.06$ | [$-4.82$, 2.59] |
| $\sigma_{pop}$ (0.5) | 0.79 | [0, 2.77] | 0.23 | [0, 0.89] |
| **(B)** | | | | |
| $\beta_{prey}$ (0.6) | 0.62 | [0.37, 0.89] | 0.299 | [0.02, 0.599] |
| $\beta_{bodysize}$ ($-0.01$) | $-0.01$ | [$-0.1$, 0.08] | $-0.004$ | [$-0.09$, 0.07] |
| $\mu_{pop}$ ($-1$) | $-1.24$ | [$-1.82$, $-0.69$] | $-0.59$ | [$-1.24$, $-0.02$] |
| $\sigma_{pop}$ (0.5) | 0.29 | [0.14, 0.43] | 0.18 | [0.06, 0.29] |

**Notes.**

$\beta_{prey}$, slope parameter for effect of number of prey items consumed; $\beta_{bodysize}$, slope parameter for effect of individual body size; $\mu_{pop}$, mean value of population random intercept term; $\sigma_{pop}$, standard deviation of population random effect.

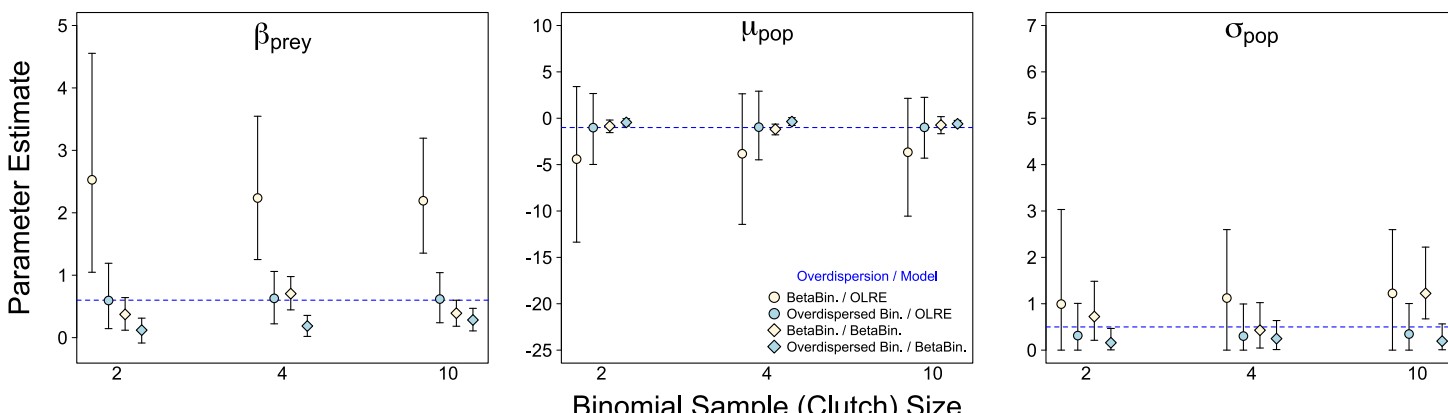

**Figure 3 Effect of Binomial sample size on accuracy of parameter estimates in the presence of overdispersion.** Parameter estimates and 95% intervals for 3 different Binomial sample sizes (clutch size) under 4 combinations of overdispersion and model type. Yellow circles, Beta-Binomial overdispersion and an observation-level random effect (OLRE) model; blue circles, overdispersed Binomial data and OLRE model; yellow diamonds, Beta-Binomial data and a Beta-Binomial model; blue diamonds, overdispersed Binomial data and a Beta-Binomial model. Beta-Binomial data were analysed using Bayesian Hierarchical Beta-Binomial mixed models and so error bars are 95% credible intervals. '$\beta_{prey}$,' slope parameter for effect of number of prey items consumed; '$\mu_{pop}$,' mean value of population random intercept term; '$\sigma_{pop}$,' standard deviation of population random effect. For all simulations $\sigma_\varepsilon$, was set to 3 for overdispersed Binomial models, and $\phi$ set to 2 for Beta-Binomial models.

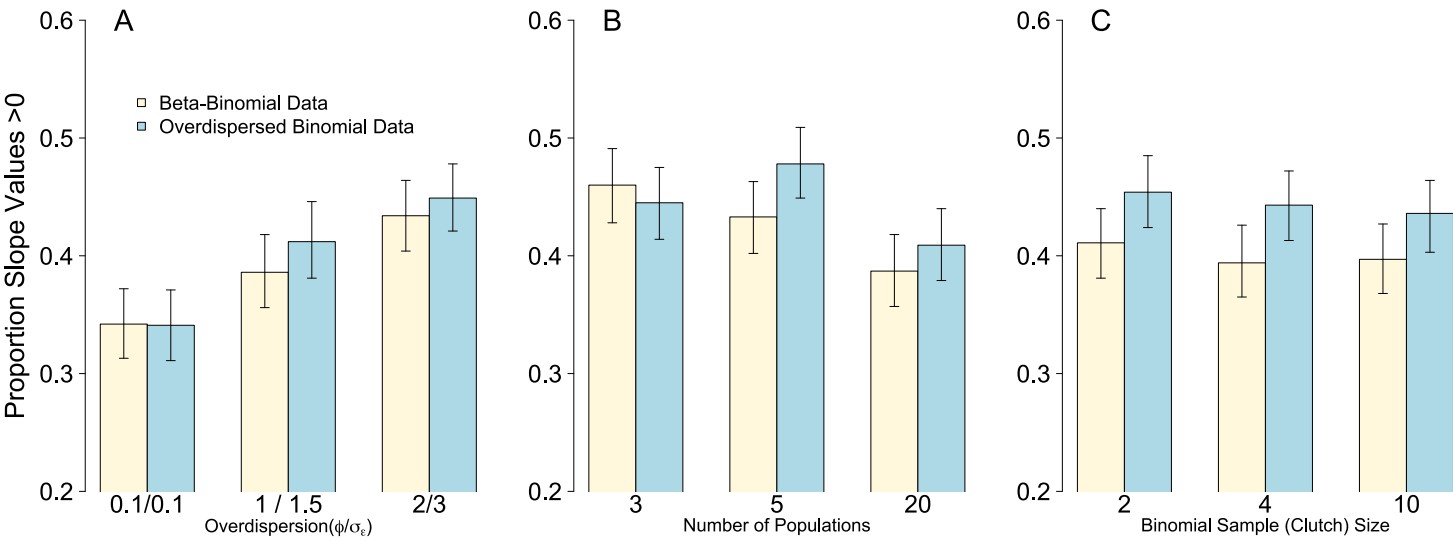

**Figure 4 Proportion of models after 1,000 simulations incorrectly estimating the weakly negative slope of the body size parameter to be positive.** Bars are mean and 95% confidence intervals following 1,000 simulations of either Beta-Binomial (yellow bars) or overdispersed Binomial (blue bars) data and analysed with OLRE models. (A) the influence of increasing levels of overdispersion; (B) the influence of increasing the replication of the random intercept term for population; (C) the influence of increasing the Binomial sample size (total clutch).

on overdispersed Binomial data leads to underestimating the values of both of $\mu_{pop}$ (middle pane, Fig. 1; Table 2C) and $\beta_{prey}$ (left pane, Fig. 1; Table 2B). Collectively these results suggest the discrepancies in parameter estimates observed between OLRE and Beta-Binomial models were not simply due to using either Bayesian or frequentist methods, but reflected a genuine difference in ability of certain models to handle certain types of overdispersion.

## DISCUSSION

Using a simulation approach, I have investigated both the ability of observation-level random effects to recover accurate parameter estimates under various degrees of overdispersion in Binomial models, and whether the performance of OLRE in mixed models is consistent across multiple types of overdispersion. In addition, I have examined how model performance changes in the presence of overdispersion when both the sample size of random effects (number of levels) and Binomial sample size is low for both OLRE and Beta-Binomial models. In general, OLRE models performed poorly when fitted to Beta-Binomial data, and this effect was particularly pronounced when the number levels of the random effect was ≤5, or the Binomial sample size was small. In all cases, increasing the number of levels of the random effect or Binomial sample size failed to remedy the bias in the estimates caused by overdispersion. Here I discuss the implications of these results for choosing OLRE as a suitable tool to model overdispersion in ecological data.

### The ability of OLRE to cope with overdispersion depends on the process generating the overdispersion

This study has shown that the ability of OLRE to recover accurate parameter estimates in overdispersed mixed models depends on the process generating the overdispersion

in the dataset. For overdispersion generated by adding random noise to the linear predictor (overdispersed Binomial data), the model recovered accurate mean estimates for slopes, intercepts and variance components at all levels of overdispersion, although the precision of the estimates declined (increasing standard errors) as overdispersion increased. Conversely, for data generated using a Beta-Binomial process, parameter estimates became increasingly more biased as overdispersion increased, leading to inflated estimates of effect size for the variables of interest. The implications of these results are that OLRE may not be a robust tool for dealing with overdispersion in Binomial mixed models because the researcher is unlikely to know which process generated the overdispersion in the first instance, meaning it is unclear if the parameter estimates are trustworthy. More worryingly, that the use of OLRE can lead to inflated effect sizes may result in researchers concluding that variables under investigation are highly influential, when in fact their effect sizes may be more modest. These patterns were not observed in the Beta-Binomial models, irrespective of data type, although the confidence intervals did also increase in concert with overdispersion. The relative utility of Beta-Binomial models over OLRE models is discussed below.

## Increasing random effect sample size increases precision, even for biased estimates

For all model/data combinations, increasing the number of populations measured from 3 to 20 greatly increased the precision of estimates for all parameters, reflected by smaller 95% confidence intervals. Importantly, this result held even when controlling for total sample size, demonstrating that higher replication *within* populations cannot compensate for fitting a population random intercept term with only 3 levels. Arguably, parameter precision was also poor when using 5 populations, especially for the variance component $\sigma_{pop}$. This corroborates the general rule of thumb that random intercept terms should ideally contain more than 5 levels in order to yield accurate estimates and good model performance (*Gelman & Hill, 2006*), especially when overdispersion is present. The key result of the random effects simulations is that when using OLRE on Beta-Binomial data, increasing sample size of the random effect does nothing to remedy the bias in slope estimates for effects such as $\beta_{prey}$. Instead, increasing the number of levels of the random effect simply makes one more certain of the accuracy of the estimates by decreasing the 95% confidence intervals, even when the mean estimate is biased. This is worrying, as well-replicated studies studying 10s to 100s of 'groups' (be they populations, genetic lines or sampling locations etc.) may recover highly precise estimates for parameters that are highly inflated with respect to their true value. In addition, it means one cannot use enormous standard errors as diagnostic evidence for suspicious behavior of OLRE (i.e., for $\beta_{prey}$ when $n$ populations $= 3$) because these are likely to change with sample size.

## Parameter estimates are largely similar irrespective of Binomial sample size

Binomial sample size had the smallest influence on model behavior of the three scenarios tested. There was some indication that parameter accuracy increased with Binomial sample

size, but these effects were modest for most data/model combinations, especially when compared to the relatively large influence of random effect sample size (see above). Such a result is intuitive, as one would expect total sample size to be more influential than simply the sample size of a single observational unit. That model precision is similar irrespective of maximum sample size is encouraging, particularly as researchers in the fields of ecology and evolution deal with an enormous range in sample sizes e.g., analyzing the hatching success of a bird that produces only 2 eggs at a time (*Harrison et al., 2013b*) or an insect that produces 100s of eggs at a time (*Tyler et al., 2013*). Collectively these results suggest that model precision need not be sacrificed if working on organisms with life history characteristics such as small clutch size or low fecundity.

## Are Beta-Binomial models more robust than OLRE models for Binomial data?

A persistent pattern in the results shown here is that OLRE perform poorly for Beta-Binomial data, yet Beta-Binomial models tend to perform well across both Beta-Binomial and overdispersed Binomial data. This does not mean that OLRE models are unsuitable for modeling overdispersion, simply that one must interpret initial model results with caution and examine the suitability of OLRE for a particular dataset . The most straightforward way to probe the robustness of OLRE model results would be to compare parameter estimates from a given OLRE model with the Beta-Binomial equivalent. There were dramatic differences in estimates for the $\beta_{prey}$ slope between models types for the Beta-Binomial data, yet very little difference for the corresponding overdispersed Binomial data, and so the comparative approach should readily identify potential problems with OLRE.

When performing model comparison, whether one uses a frequentist statistical package, or codes the model manually in a Bayesian framework in *JAGS* appears to be a matter of preference, as these two approaches yielded similar results in the current study. Although Bayesian methods do have several advantages over frequentist methods (e.g., *Ellison, 2004*; *Kery, 2010*), in many cases they recover similar parameter estimates to frequentist models when uninformative priors are used (e.g., *Kery, 2010*). In support of this, sensitivity analyses presented in this paper suggest that model results are similar whether one uses a Bayesian or a frequentist Beta-Binomial model, meaning choice of model rather than of statistical philosophy is the more important driver here. However, I would caution that I only repeated the frequentist Beta-Binomial simulations for a limited subset of cases (high overdispersion, 10 populations, 20 individuals per population), and so it should not be assumed that frequentist and Bayesian approaches would agree in every case (see also examples in *Kery, 2010*). A final caveat for these results is that the data generated in this study are 'ideal' in so far as they are perfectly balanced across populations (identical numbers of individuals per population). In reality this is unlikely to hold, as ecological datasets often contain poorly represented groups with far lower sample sizes than others e.g., years with poor breeding success and limited data on clutch size (e.g., *Harrison et al., 2011*; *Harrison et al., 2013b*). How imbalance in sample size affects model estimates in the presence of overdispersion for two models with identical number of levels of the random intercept term warrants further investigation.

Superficially, it appears that Beta-Binomial models perform better than OLRE models in most cases, and so a natural inclination would be to simply use Beta-Binomial models for any kind of overdispersed Binomial data. However, results from this study indicated that Beta-Binomial models can underestimate slope values (e.g., the value of $\beta_{prey}$), whereas the corresponding OLRE model does not. Thus, Beta-Binomial models do not universally outperform OLRE models, and one should not sacrifice OLRE from the set of tools available to deal with overdispersed Binomial data. Both Beta-Binomial and OLRE models estimate an overdispersion parameter that can reveal a biological cause underlying aggregation/non-independence of probabilities in the dataset (e.g., Beta-Binomial models, *Hilgenboecker et al., 2008*; OLRE models, *Elston et al., 2001*, and the size of the aggregation parameter is informative and comparable across studies (Richards, 2008)). Beta-binomial models are frequently employed in the ecological literature to model non-independence among probabilities (e.g., *Hughes & Madden, 1993*; *Lee & Nelder, 1996*; Clark, 2003; Richards, 2008) and may be less prone to overfitting than the corresponding OLRE models, which may explain why OLRE models performed poorly for Beta-Binomial data. Indeed, the dispersion parameter of models containing OLRE frequently collapses to <0.5 (data not shown), suggesting the addition of OLRE replaces overdispersion with underdispersion, which can be equally as problematic (*Zuur et al., 2009*).

## Summary

Observation-level random effects provide a simple means to control overdispersion that can be easily implemented in mixed effects model packages. However, it is clear that their use may not be appropriate in all cases. Results from models containing OLRE should be carefully inspected, and where possible corroborative evidence should be sought from alternative modeling approaches such as (Bayesian) Hierarchical Beta-Binomial models to quantify agreement between parameter estimates and ensure the conclusions drawn from such analyses are robust. Finally, one should avoid fitting random intercept terms to models when the random term contain <5 levels, especially in the presence of overdispersion, as parameter estimates become unreliable irrespective of modeling approach. One should also interpret model results with caution when the random effect sample size is large (e.g., >20), because models with OLRE can yield inaccurate but precise (small confidence intervals) slope estimates under certain scenarios that may give the false impression of that model having performed well.

### Funding

This work was funded by a Research Fellowship awarded to XH by the Zoological Society of London, and a BES Research Grant (Grant Number 4720/5758). The funders had no role in study design, data collection and analysis, decision to publish, or preparation of the manuscript.

## Grant Disclosures

The following grant information was disclosed by the author:
Zoological Society of London.
BES Research Grant: 4720/5758.

## Competing Interests

The author declares there are no competing interests.

## Author Contributions

- Xavier A. Harrison conceived and designed the experiments, performed the experiments, analyzed the data, contributed reagents/materials/analysis tools, wrote the paper, prepared figures and/or tables, reviewed drafts of the paper.

## Data Deposition

The following information was supplied regarding the deposition of related data:

R Code for the analyses in this paper are deposited on FigShare: http://dx.doi.org/10.6084/m9.figshare.1450680.

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
