# Peer review of "A comparison of observation-level random effect and Beta-Binomial models for modelling overdispersion in Binomial data in ecology & evolution"

_PeerJ, doi:10.7717/peerj.1114_

## Round 0.1 · original submission · Major Revisions

Both referees agree that your paper might represent an interesting contribution, but they also raise important issues (and not just with respect to notations and presentations - note that these issues are important, you cannot replicate studies like yours if notations are not precise). Considering different sample sizes would also strengthen your paper.

Reviewer 1 ·

Basic reporting

The paper deals with the interesting problem of overdispersion on binomial data. The author presents some simulations studies to verify the suitability of binomial random effects models and beta-binomial models in the situation of overdispersion. The fitting of binomial random effects is carrying out using the R software and the well-known lme4 package. The package runjags was used to fit the beta-binomial models, under the Bayesian framework. In spite of interesting subject the paper has a lot of problems that should be solved to improve its quality.

1) Notation is terrible using non standard mathematical symbols to describe mathematical computations for example in equation (2) the author used mean.pi it should be the expectation of the random variable $h_i$ whose index are not described by the author and using standard mathematical notation should be $E(h_i)$. This kind of non standard notation becomes the paper really hard to read and unclear.

2) Almost all equations are not well described with index non defined or inconsistent, for example the index appears on left-side but not on right-side of equations. Example this can be seen on equation (2) where the index $j$ appears on the right-side but not on the left-side.

3) The R code is impossible to read, the components and arguments are not described.

4) Capital letters are used in a non standardized way, for example beta-binomial or Beta-Binomial the same for binomial or Binomial.

5) The article is quite confused with a lot of small subsections and equations repeated, for example equation (2) appears on page 2 and 5.

6) The discussion section is confused, is quite hard following the author. The figure's caption does not make sense. For example, in Figure A the parameter $\beta_{prey}$ has the value $-0.1$, but its values appears always bigger than $1$.

I suggest a comprehensive revision in terms of notation, format and readability of the article with the objective to become the article easier to read.

Experimental design

The simulation study design is limited, since the author used only one sample size (20) and numbers of individuals (200). In general random effect models are hard to fit by maximum likelihood and the sample size and numbers of individuals are always important features to consider in simulation studies. Furthermore, is well-known in the literature that binomial random effects models are even more difficult to fit and the number of trials is a critical point for the estimation algorithm. In the authors notation the number of binomial trials is denoted by $c$ but I could not find where this number was specified by the author.

I suggest a clearer description the number of trials used in the simulation study. Furthermore, I suggest that a more comprehensive simulation study considering the effect of sample size, number of individuals and binomial trials be considered.

Validity of the findings

The author does not describe the prior specification used in the Bayesian beta-binomial model. The author does not provide any sensibility analysis to the prior distribution used in the Bayesian beta-binomial model. The author assumes that the maximum likelihood estimator is able to recover the model parameters in a suitable way. But for binomial random effects models it may be not true, since the marginal likelihood is an intractable integral that should be solved numerically and the method used to do it may be an important effect on the final estimates. This effect was ignored by the author on his discussion.

I suggest to include a clear description of the prior distributions used in the Bayesian models and a sensibility analysis to verify the impact these distributions on the final estimates.

Additional comments

The estimation of binomial mixed models is not a easy task, when dealing with simulation studies the fitting algorithm used to estimate the model parameters is very important in the final result. I think you should at least have a look in this paper http://xxx.tau.ac.il/abs/1503.07307 for a discussion these problems and take care about your conclusions. This paper deals with Bayesian inference, but it is quite similar for likelihood methods.

Reviewer 2 ·

Basic reporting

I think the paper is reasonably easy to understand and has fairly clear objectives and findings. Some more critical detailed comments are included in the sections below.

The topic is essentially statistical, and the author is probably not a statistician. Hence there is a lot of statistical literature that is ignored. For example, the comparison between the two types of model discussed here is discussed in the paper

Williams, D.A. (1982) Extra-binomial variation in logistic linear models. Journal of the Royal Statistical Society, Series C (Applied Statistics), 31, 144-148,

which currently has 745 citations in Google Scholar.

Experimental design

The structure of the paper seemed a little odd to me, in that overdispersed data are generated in two ways (by adding random effects on the logistic scale or via a beta-binomial distribution) and analyses are presented that parallel these (using OLRE or via a beta-binomial model). What I thought was odd was that the main focus was on OLRE, with beta-binomial analysis added more as an afterthought (see e.g. last sentence of abstract). Why not treat them more symmetrically?

The basic approach of simulating data so that estimates can be compared with known parameter values is a common one. The simulations here are not particularly extensive, since they relate to the rather specific model (2), where there are two regression coefficients with opposite signs, one being a weak effect, the other stronger. Also, one might argue that the presence of a random intercept term in the model rather clouds the issue of overdispersion and it might have been nice to see at least some simulations from a model with a constant intercept term.

The mathematical description of the model could be improved. For example:

The symbol i is introduced on line 52, but it is not properly explained that this represents an individual until line 65.

The binomial distribution is written ‘Binomial(pi,ci)’ whereas the standard notation is to put the index (ci) before the probability.

The brackets in the denominator of the RHS of eqn (3) are redundant [and the RHS could be written more simply as 1/(1+exp(-mean.pi)].

In eqn (4) the normal distribution is defined in terms of its mean and standard deviation, but the standard notation for a normal distribution uses mean and variance.

In lines 71/72 the R code doesn’t fit on the line and can’t be read properly. Similarly later, lines 100-102, line 173 and line 176.

Line 81/82 ‘indicating a significant degree of overdispersion’. This would depend on the degrees of freedom. Do you just mean a ‘considerable degree’? Maybe better still would just be to replace this by ‘If the data exhibit overdispersion, we can adjust our model to take this into account, …’

The notation has gone wrong in eqns (7)-(10). What appears as pi in (8) and (9) should I suspect be mean.pi and probi should just be pi.

In the description of the data generation (lines 152-159) the value of h should be specified.

On line 169, I would suggest replacing ‘Each dataset was generated 1000 times …’ by ‘One thousand data sets were simulated …’

In the beta-binomial analysis (line 182 onwards) it is not explained why only two combinations of parameter values are considered here. I assume this is because of computation time?

Line 185, ‘few frequentist mixed model packages in R permit the fitting of beta-binomial models.’ Are there any? If so, could they be used.

Validity of the findings

The RESULTS section is succinct, but hard to follow because of multiple errors in table and figure captions. For example in Table 1 caption, the slope quoted for bodysize should be for prey. (And it would be helpful to give the true value for με in the table caption). The subscripts are missing in the row column labels of the table.

The confusion of prey and bodysize coefficients continues in the Figure captions. Also μ_pop seems to be mixed up with σ_pop.

Line 224, ‘the weak but significant effect of body size’. Again this looks like a non-statistical usage of the word ‘significant’, which is best avoided.

The findings are not overall very surprising – that different methods of analysis may be more or less successful depending on the model that generated the data. What is somewhat surprising is the relatively good performance of the beta-binomial model across different generative models. I think the author is right to emphasise in the Discussion that this is probably not due to the fact that this analysis is Bayesian but the others are frequentist (though this is a confounding factor). But this leaves the question of why the beta-binomial model appears to have broader applicability than the OLRE model. Some insight into this would strengthen the paper.

---

## Round 0.2 · Minor Revisions

You have answered most referees comments adequately but there are minor issues that need to be addressed:

L 54: remove underline «from its total clutch»
l. 67: drawn from
l. 70: body size
l. 71: why “mean probability” – p_i is the probability for each individual. Mean p_i might also be a confusing notation since here it is the constant logit(p_i). It could be a mean(logit(p)) only for the latter model with extra individual variation (and even that could be misused since the logit(mean(p_i)) is not the same as mean(logit(p_i))). If you want ecologists to read through the paper you have to use a precise terminology.
l. 98: same comment here, mean.p_i is a bit confusing, it is the logit(p_i), not a mean of anything.
l. 117: parenthesis missing after “2001)”
l. 136: why a prime for the beta.p_i?
l. 223: replace & by and
l. 270: remove “,” after “data.”
l. 282: replace “incredibly” by “highly” or something equivalent.
Figure 4 did not look ok (x axis and legend).
Table 2 legend “parameter estimates”, and “body size” (instead of bodysize)

---

## Round 0.3 · accepted · Accept

The final minor changes have been made.